# The impact of Duration of Untreated Psychosis on functioning and quality of life over one year of Coordinated Specialty Care (CSC)

Hadar Hazan[1]*, Sümeyra N. Tayfur[1], Bin Zhou[2], Fangyong Li[2], Toni Gibbs-Dean[1], Sneha Karmani[1], Emily Kline[3], Maria Ferrara[4], Silvia Corbera[5], Sarah Riley[1], Laura A. Yoviene Sykes[1], Cenk Tek[1], Matcheri S. Keshavan[3], Vinod H. Srihari[1]

1 Department of Psychiatry, Yale University School of Medicine, and Program for Specialized Treatment Early in Psychosis (STEP), New Haven, Connecticut, United States of America, 2 Yale Center for Analytical Sciences, Yale School of Public Health, New Haven, Connecticut, United States of America, 3 Department of Psychiatry, Harvard Medical School, Beth Israel Deaconess Medical Center, and Massachusetts Mental Health Center, Boston, Massachusetts, United States of America, 4 University of Ferrara, Ferrara, Italy, 5 Department of Psychological Science, Central Connecticut State University, New Britain, Connecticut, United States of America

* hadar.hazan@yale.edu

**Data Availability Statement:** The data is securely stored on a Yale server, managed and maintained by the Yale Center for Analytical Sciences (YCAS),

## Abstract

### Background

This study examined the relationship between the Duration of Untreated Psychosis (DUP) and functional outcomes at baseline, 6 months, and 12 months after admission to Coordinated Specialty Care (CSC).

### Methods

A total of 246 participants from two U.S. public-sector CSC programs were categorized into Low and High DUP groups using two criteria: (i) a median split of the DUP distribution and (ii) the World Health Organization (WHO) aspirational standard of 3 months. Changes in global functioning (GAF), social functioning (GF: Social), and occupational functioning (GF: Role), symptom severity (PANSS), and quality of life (QoL) were assessed using a Linear Mixed Model Repeated Measures (MMRM) analysis across the three time points. A Generalized Linear Model (GLM) with a logit link function was applied to analyze binary outcomes, specifically the status of being Neither in the Labor Force, Education, or Training (NLFET). Both models adjusted for time and site as covariates and used an unstructured variance-covariance matrix to account for within-subject correlations in repeated measures. The difference-in-differences method was employed to estimate the impact of DUP on outcomes over time, with results reported as least square means for continuous outcomes, odds ratios for binary outcomes, and 95% confidence intervals (CI) for both Low and High DUP groups.

in compliance with institutional standards for data security and confidentiality. This server is approved for the storage of 'High Risk' data, including Patient Health Information (PHI), and is accessible to YCAS staff. Researchers who meet the criteria for confidential data access may request it through the Yale IRB. For access inquiries, please contact Alicia Lakomski, Senior Administrative Assistant at YCAS, at alicia.lakomski@yale.edu.

**Funding:** This work was supported by National Institutes of Health (R01MH103831) and the Gustavus and Louise Pfeiffer Research Foundation. The funding sources had no role in the design and conduct of the study; collection, management, analysis, and interpretation of the data; preparation, review, or approval of the manuscript; and decision to submit the manuscript for publication. This work was also funded by the State of Connecticut, Department of Mental Health and Addiction Services, but this publication does not express the views of the Department of Mental Health and Addition Services or the State of Connecticut. The views and opinions expressed are those of the authors.

**Competing interests:** The authors have declared that no competing interests exist.

## Results

No significant differences were observed between the Low and High DUP groups at baseline. By 6 months, participants in the Low DUP group (DUP < 3 months) exhibited significantly greater improvements (reduction) in NLFET status (3-month OR = 3.25, p = 0.03; median split OR = 2.25, p = 0.03) and global functioning, GF: Role, and GF: Social. By 12 months, the Low DUP group continued to show significantly greater NLFET status improvement (3-month OR = 3.59, p = 0.03; median split OR = 3.40, p = 0.0032). Improvements in global functioning were sustained at 12 months, while social and occupational functioning gains were not. No significant differences were observed between groups for quality of life or symptom severity over time.

## Conclusion

Shorter DUP is linked to more rapid functional recovery within the first year after CSC admission.

## Introduction

Individuals with schizophrenia spectrum disorders who experience long delays between psychosis onset and treatment, measured as the Duration of Untreated Psychosis (DUP), often face a range of adverse functional outcomes [1]. Psychotic illnesses typically onset in a developmentally vulnerable period spanning late adolescence and early adulthood, when prolonged DUP can negatively impact several measures of functioning, including social functioning [2], likelihood of maintaining full-time employment, global functioning [3], and quality of life [4].

Relatedly, there is growing worldwide concern for youth who are Not Engaged in Education, Employment, or Training (NEET). Samples of recent onset schizophrenia spectrum disorders or First Episode Psychosis [5] (FEP) exhibit high levels of NEET (39.2%-41.7%), and this is associated with longer DUP [6–8]. However, variable operationalizations of NEET have limited comparison and accumulation of knowledge across studies. Recognizing this, the International Labor Organization introduced the more developmentally appropriate category of NLFET (Neither in the Labor force nor in Education or Training) to include within the labor force those unemployed youth who are actively looking for work [9].

Specialty team-based clinical services (termed Coordinated Specialty Care or CSC in the US) for early course schizophrenia spectrum disorders have raised expectations for functional recovery [10]. CSCs provide comprehensive medication and psychosocial care, including support for education and employment, that is specifically adapted to the needs of younger patients and their families. Experimental studies of such services have demonstrated effectiveness across varied international settings [11]. However, in one of the largest trials conducted in the U.S., prolonged DUP was found to moderate the effectiveness of CSC versus usual care [12]. The effect size for improvement in the primary outcome of quality of life fell from 0.54 to 0.07 for those with longer DUP (i.e., group above the median split of 74 weeks) in this nationally representative sample drawn from 34 community-based clinics across 21 U.S. states [12]. This was an important reminder that timing of treatment is an important determinant of functional outcomes.

While several observational studies had previously corroborated the adverse prognostic impact of prolonged DUP, inevitable variations in care quality and newer generations of care

models have limited how much we can infer from this literature about the magnitude and time course of this effect of DUP on the outcomes of modern CSCs. This secondary analysis of data from a trial of early detection leverages careful measurement of DUP and outcomes in two established and comparable modern U.S. CSCs. We specifically examined the relationship between DUP and measures of community functioning over the first year of CSC. We hypothesized that lower DUP [13] would predict better functional outcomes and that this effect would be discernible within the first year of treatment.

## Methods

### Setting and study design

Subjects for this analysis were drawn from consecutive admissions over 5 years (Feb 1, 2014-Jan 31, 2019) to two Coordinated Specialty Care (CSC) clinics: Prevention and Recovery in Early Psychosis (PREP®) [14] and Specialized Treatment Early in Psychosis (STEP) [15]. After a shared baseline year, an early detection campaign was implemented for 4 years across the catchment of STEP (Greater New Haven, Connecticut), while PREP® continued usual detection (metropolitan Boston, Massachusetts) [16]. A study protocol detailing the design was published before implementation. All participants provided written informed consent, and all study procedures were approved and monitored by the Institutional Review Boards of Yale University, Beth Israel Deaconess Medical Center, and the Massachusetts Department of Mental Health [16].

### Measures

**Duration of Untreated Psychosis (DUP).** The Structured Interview for Psychosis-risk Syndromes (SIPS) [17] was used to establish psychosis onset, operationalized with the Presence of Psychotic Syndrome (POPS) criteria. Psychosis was determined to be present when ratings on items P1-P5 of SIPS were at least 6, with at least one of these symptoms occurring over one month for a minimum of one hour per day for a minimum average of 4 days per week or leading to serious disorganization or dangerousness. The POPS date was determined with input from all available stakeholders involved in the pathway to care. The latter was assessed with a structured questionnaire [18]. DUP was measured as the time in days from psychosis onset to enrollment in a CSC (STEP or PREP) [16].

**Outcome measures.** Neither in the labor force nor in Education or Training (NLFET) status was assessed using the Bureau of Labor Statistics (BLS) linked categorization form [19]. BLS standardized codes are used in various databases to track employment trends, unemployment rates, wages, and other economic indicators (S1 Fig). The information collected on the BLS-linked categorization form was used to generate a numerical BLS code that classifies the patient's employment status.

The Global Assessment of Functioning Scale (GAF) [20] (integrated within the SIPS) was used to measure overall levels of current and recent (12 months prior) psychological, social, and occupational functioning. Assessors computed omnibus scores (from a low of 1 to a high of 100) that combined ratings for one or more areas of impairment across these three domains. Social and role functioning was assessed with the GF: Social and GF: Role scales to delineate functioning in specific social and occupational roles, respectively [20]. Patients were assessed for current and recent (lowest and highest levels 12 months prior) functioning levels (ranging from a low of 1 to a high of 10).

The Positive and Negative Symptom Scale (PANSS) [21] was used to assess psychosis symptom severity, including scores for General, Positive, and Negative symptom domains. The Premorbid Adjustment Scale (PAS) [22] was used to measure functioning for the childhood

epoch (age 5–12 years) that precedes the typical onset of prodromal symptoms. The Hein-richs-Carpenter Quality of Life Scale (QLS) [23] was used to measure the overall sense of pur-pose, motivation, emotional and social interaction, role functioning, and engagement in regular activities. The scale includes 21 items rated from a semi-structured interview with total scores ranging from 0 (worst) to 126 (best). The Wide Range Achievement Test 4 (WRAT4) [24] reading section was used to provide an estimate of premorbid IQ.

**Reliability procedures.** Raters at each site were trained using a set of common vignettes developed from historical cases at each site. Remote and in-person meetings used a mix of written and video vignettes for inter-rater reliability assessments that included SIPS and all the functional measures used in this analysis. After achieving adequate reliability, raters repeated this exercise after every 20 new subjects recruited at each site to ensure ongoing reliability monitoring. Annual meetings between the sites were held to reduce drift and increase reliability.

Further procedures to ensure accuracy in the measurement of DUP included weekly calls with staff across both sites. Each enrolled subject was presented within a structured format. Investigators across both sites oversaw consensual confidence ratings of DUP based on a rubric that considered the quality of information from the patient and collateral sources (care-givers, medical records). Cases with ambiguous ratings were revisited after one month to allow time for additional collateral or improved assessment of more symptomatically stabilized patients.

## Statistical analysis

Classification into Low versus High DUP was based on (i) a median split of the distribution (High DUP, n = 123; Low DUP, n = 123) and (ii) the World Health Organization (WHO)/ International Early Psychosis Association (IEPA) aspirational standard of 12 weeks (High DUP, n = 173; Low DUP, n = 73) [25]. Linear Mixed Model Repeated measures (MMRM) analysis was utilized to evaluate change in continuous outcomes (functioning, symptom sever-ity, quality of life), and Generalized Linear Model (GLM) with logit link function was utilized to evaluate change in binary outcomes (for employment, being in school, or job hunting), in which visit times (baseline, 6-month and 12-month) and site were adjusted as covariates for all outcomes, with an unstructured variance-covariance matrix specified to account for within-subject correlation of repeatedly measured values during periods. Least square means for con-tinuous outcomes, odds ratios for binary outcomes, and confidence intervals (CI) were reported at each time point by Low and High DUP.

GAF scores were adjusted based on pre-morbid IQ, as measured by the Wide Range Achievement Test, and childhood scores from the Premorbid Adjustment Scale (PAS). Symptom severity adjustments were made using the baseline scores as a covariate. Quality of life and NLFET status adjustments accounted for site and visit time without incorporat-ing additional covariates. The Difference-in-Differences (DiD) method was employed to estimate the causal effect of Low DUP on changes in outcomes from baseline to 6 and 12 months. This approach compared temporal changes in outcomes between the Low DUP and High DUP groups, controlling for baseline differences and shared time-related trends. Specifically, DiD calculated the change in outcomes—functioning, symptom severity, and quality of life—for each group at admission, 6 and 12 months post-admission, and then compared the changes between the two groups. This method effectively controls for any baseline imbalances and identifies whether the Low DUP group demonstrated greater improvements over time than the High DUP group, thereby suggesting a causal relationship between DUP and improved outcomes.

## Results

### Participants

STEP and PREP® recruited comparable, diverse, and regionally representative samples [26]. The participants were young, with a median age of 21.6 years (IQR:19.7 to 24.5), and predominantly male (71%, $n$ = 174). The sample exhibited substantial racial diversity, with 45% (n = 110) identifying as Black, 32% (n = 79) as White, and 15% (n = 37) as multi-racial. Educational attainment was generally low, with 85% (n = 210) having completed only high school. Most participants (96%, n = 236) were single and had never married.

The average WRAT score (Mean = 99.71, $SD$ = 16.30) was close to the general population average. The PANSS total score (Mean = 72.1, $SD$ = 16.49) indicated moderate to severe symptoms, while the GAF score (Mean = 33.6, $SD$ = 11.2) reflected serious impairment in functioning. The QoL score (median = 59.0, IQR: 47.0–74.0) suggested a moderate level of impairment. About a third of the sample at CSC admission (34%, $n$ = 84) were neither employed nor engaged in education or training (NLFET) (Table 1).

### Differences in functional outcomes between Low and High DUP groups

The median DUP for the sample was 228 days (IQR: 67 to 533). Descriptive statistics summarized the outcome variables for Low and High DUP groups at baseline, 6 months, and 12 months using two cutoffs: the median split and the WHO 3-month standard (S1 Table). The relationships between DUP and outcome measures, including NLFET status, GAF scores, symptom severity, and quality of life (QoL), are illustrated in figures divided into two panels, representing each cutoff criterion.

### Community functioning

**NLFET status (Neither in the labor force nor Education or Training).**   At baseline, there were no significant differences in NLFET status between the Low DUP and High DUP groups for both the 3-month (OR = 1.32, 95% CI: 0.69 to 2.51, p = 0.40) and the median split cutoffs (OR = 1.24, 95% CI: 0.71 to 2.16, p = 0.45). By 6 months, the High DUP group demonstrated significantly higher odds of NLFET status compared to the Low DUP group in both 3-month (OR = 3.25, 95% CI: 1.11 to 9.54, p = 0.03) and median split cutoffs (OR = 2.25, 95% CI: 1.08 to 4.71, p = 0.03). This trend continued at 12 months, with the High DUP group maintaining significantly higher odds of NLFET status (3-month cutoff: OR = 3.59, 95% CI: 1.10 to 11.73, p = 0.03; median split: OR = 3.40, 95% CI: 1.51 to 7.68, p = 0.0032). These findings indicate that, while both groups improved over time, the High DUP group consistently showed a more pronounced likelihood of remaining in NLFET status compared to the Low DUP group (Fig 1).

**Global assessment of functioning (GAF).**   The Difference-in-Difference (DiD) analysis showed that individuals in the Low DUP group improved significantly more in their GAF scores during the first 6 months compared to those in the High DUP group. Specifically, the Low DUP group had a much larger increase in GAF scores (3-month cutoff: DiD = -11.2, 95% CI: -15.7 to -6.7, p < 0.0001; median split: DiD = -6.8, 95% CI: -11.0 to -2.6, p = 0.0016).

However, from 6 to 12 months, the rate of improvement between the two groups leveled out, as there were no significant differences in the change in GAF scores (3-month cutoff: DiD = 3.1, 95% CI: -1.5 to 7.8, p = 0.18; median split: DiD = 0.9, 95% CI: -3.3 to 5.1, p = 0.66). This suggests that the initial advantage for the Low DUP group was most pronounced in the first 6 months of treatment, and thereafter, both groups improved at similar rates (Fig 2).

**Table 1. Sample characteristics at CSC admission.**

|  | STEP & PREP |
| --- | --- |
|  | (N = 246) |
| **Age**, years, Median (IQR) | 21.6 (19.7–24.5) |
| DUP total, days, median (IQR) | 228 (67–533) |
| **Sex at birth** |  |
| Female | 72 (29%) |
| Male | 174 (71%) |
| **Race** |  |
| White | 79 (32%) |
| Black | 110 (45%) |
| Interracial | 37 (15%) |
| Other | 20 (8%) |
| **Education** |  |
| College and above | 36 (15%) |
| High school | 210 (85%) |
| **Marital** |  |
| Single, never married | 236 (96%) |
| Other | 10 (4%) |
| **WRAT (sum score)** |  |
| Mean (SD) | 99.71 (16.30) |
| Median (Range) | 99.0 (64.0–145.0) |
| **PAS** |  |
| Mean (SD) | 1.97 (0.25)s |
| Median (Range) | 2.0 (0.0–2.0) |
| **PANSS total** |  |
| Mean (SD) | 63.15 (17.68) |
| Median (Range) | 62.0 (30.0–152.0) |
| **GAF** |  |
| Mean (SD) | 43.62 (15.41) |
| Median (Range) | 43.0 (8.0–90.0) |
| **QOL** |  |
| Mean (SD) | 68.51 (21.35) |
| Median (Range) | 66.0 (17.0–122.0) |
| **NLFET** (n) | 34.15% (84) |

Note. WRAT: Wide Range Achievement Test (sum score); PAS: Premorbid Adjustment Scale; PANSS total: Positive and Negative Symptom Scales Total Score; GAF: Global Assessment of Functioning; QOL: Quality of Life; NLFET: Neither in the Labour Force nor in Education or Training (BLS = N3 Not in the labor force—has not looked for work in past 4 weeks, does not consider self available to work)

**GF: Role.** The DiD analysis showed a significant improvement in GF: Role scores from baseline to 6 months, with the Low DUP group improving more than the High DUP group in both cutoff models (3-month cutoff: DiD = -0.93, 95% CI: -1.60 to -0.25, p = 0.01; median split: DiD = -0.80, 95% CI: -1.40 to -0.20, p = 0.01). However, between 6 and 12 months, there were no significant differences in improvement between the groups (3-month cutoff: DiD = 0.62, 95% CI: -0.06 to 1.29, p = 0.06; median split: DiD = 0.6, 95% CI: -0.0 to 1.2, p = 0.05). This suggests that both groups stabilized and showed no further improvement during this period (Fig 2).

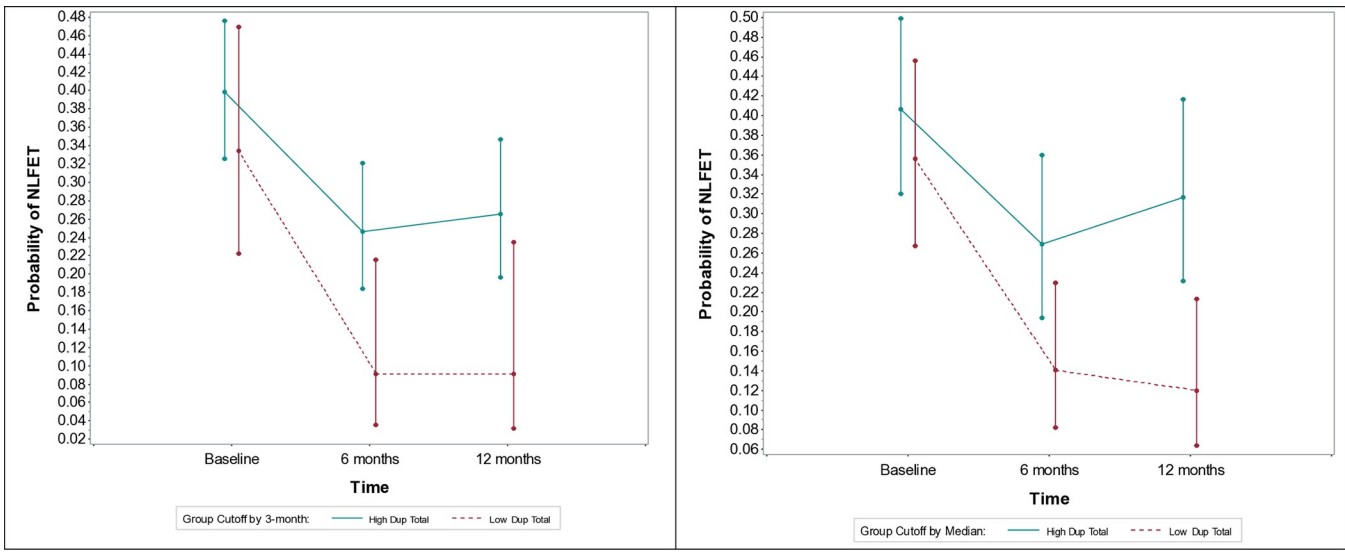

**Fig 1. Probability of Neither in the Labor Force, Education, or Training Low vs High duration of untreated psychosis (3m vs. median cutoff) groups at baseline, 6, and 12-month follow-up.**

**GF: Social.** The DiD analysis showed a significant improvement in GF: Social scores for the Low DUP group from baseline to 6 months, with better outcomes compared to the High DUP group in both cutoff models (3-month cutoff: DiD = -0.60, 95% CI: -1.07 to -0.13, p = 0.01; median split: DiD = -0.40, 95% CI: -0.80 to 0.00, p = 0.06). Between 6 and 12 months, there were no significant differences in improvement between the groups (3-month cutoff: DiD = 0.18, 95% CI: -0.29 to 0.65, p = 0.45; median split: DiD = 0.20, 95% CI: -0.20 to 0.70, p = 0.27), indicating that social functioning remained stable during this period (Fig 2).

**Quality of life.** The DiD analysis showed no significant differences in Quality of Life (QoL) improvements between the groups from baseline to 6 months for both cutoff models

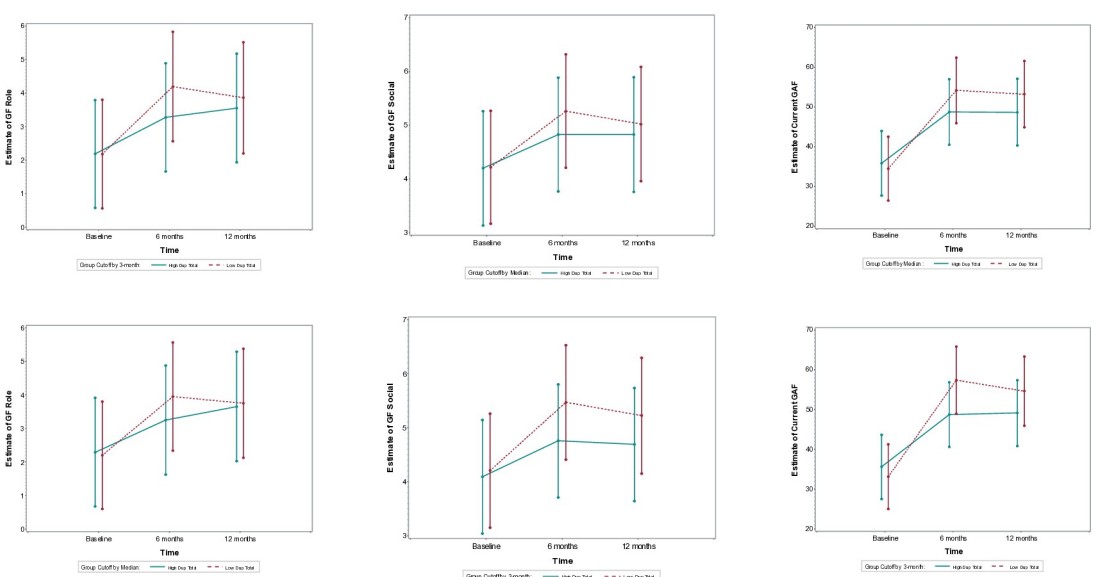

**Fig 2. Functioning outcomes for Low vs High duration of untreated psychosis (3m cutoff right vs. median cutoff) groups at baseline, 6, and 12-month follow-up.**

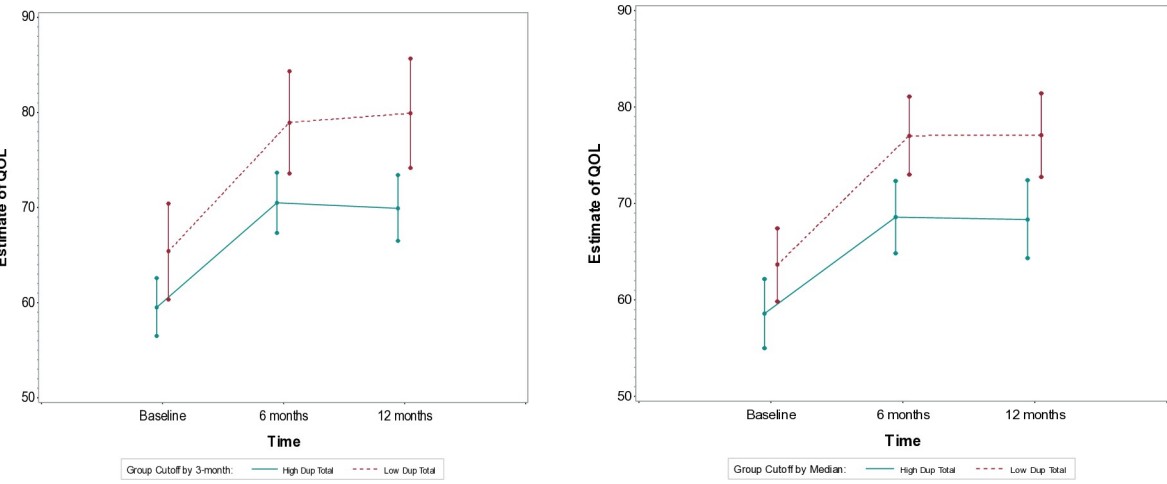

**Fig 3. Estimate of quality-of-life for Low vs High duration of untreated psychosis (3m vs. median cut off) groups at baseline, 6, and 12-month follow-ups.**

(3-month cutoff: DiD = -2.6, 95% CI: -8.6 to 3.5, p = 0.40; median split: DiD = -3.4, 95% CI: -8.8 to 2.1, p = 0.22). Similarly, no significant differences were found from 6 to 12 months (3-month cutoff: DiD = -1.5, 95% CI: -7.6 to 4.5, p = 0.62; median split: DiD = -0.3, 95% CI: -5.8 to 5.2, p = 0.92). These results suggest that neither group showed a clear advantage in improving QoL over the study period (Fig 3).

## Symptom severity (PANSS scores)

The DiD analysis for PANSS scores found no significant differences between Low and High DUP groups in the rate of change from 6 to 12 months across all symptom domains: positive symptoms (3-month cutoff: DiD = 0.4, 95% CI: -1.2 to 2.1, p = 0.61; median split: DiD = 0.2, 95% CI: -1.4 to 1.7, p = 0.82), negative symptoms (3-month cutoff: DiD = -0.2, 95% CI: -2.2 to 1.8, p = 0.83; median split: DiD = -0.4, 95% CI: -2.3 to 1.4, p = 0.64), general psychopathology (3-month cutoff: DiD = 1.6, 95% CI: -1.1 to 4.2, p = 0.24; median split: DiD = 0.4, 95% CI: -2.0 to 2.7, p = 0.77), and total symptom scores (3-month cutoff: DiD = 1.9, 95% CI: -3.0 to 6.7, p = 0.45; median split: DiD = 0.2, 95% CI: -4.3 to 4.6, p = 0.94). These findings suggest no differential symptom improvement or worsening between the groups from 6 to 12 months (Fig 4).

## Discussion

We examined the relationship between Duration of Untreated Psychosis (DUP) and functional outcomes during the first year following admission to Coordinated Specialty Care (CSC). Our findings suggest that shorter DUP (defined as below the median split or less than three months) is associated with greater improvements in functioning, as evidenced by a significant reduction in the percentage of participants classified as Neither in the Labor Force, Education, or Training (NLFET) and increase in Global Assessment of Functioning (GAF) scores at both 6- and 12-month follow-ups. Notably, shorter DUP was linked to better occupational functioning (GF: Role) at 6 months, but the High DUP group caught up by 12 months. Contrary to our initial expectations, quality of life improvements were not significantly different between the Low and High DUP groups. Additionally, no significant associations were found between DUP and psychosis symptom severity across any of the PANSS domains.

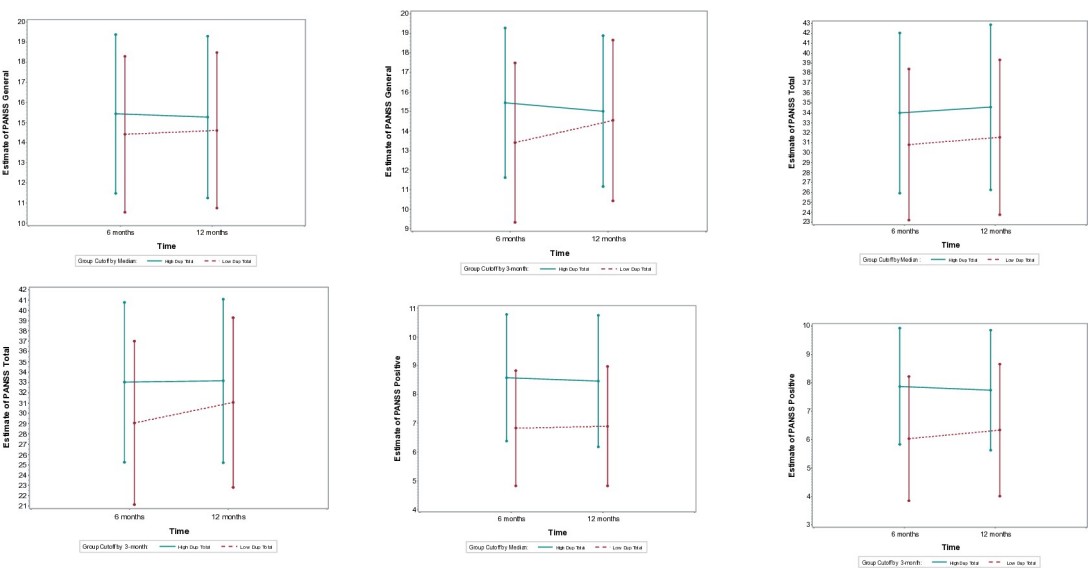

**Fig 4. Estimate of Positive and Negative Symptom Scale scores for Low vs High Low vs High duration of untreated psychosis (3m VS median cut off) groups at baseline, 6, and 12-month follow-ups.**

Our main finding—the link between longer DUP and a higher likelihood of NLFET status —aligns with prior research demonstrating that prolonged untreated psychosis is associated with decreased vocational engagement [6]. NLFET, as used by the U.S. Bureau of Labor Statistics, offers a transparent and age appropriate measure of functional capacity and social integration. While the more commonly used NEET (Not Engaged in Education, Employment, or Training) includes youth who may have the capacity but choose not to engage in developing their human capital [27], NLFET excludes those who are currently unemployed but actively looking for work or educational opportunities. Thus, being classified as NLFET reflects a significant disconnection from mainstream society, and identifies a group who may face specific illness-related barriers and are thus worthy of study, and targeted intervention. Lack of engagement in employment and education, particularly not even considering oneself as available for these activities, presents a considerable barrier to recovery in those with severe mental illness [28, 29]. In our sample, individuals with shorter delays to CSC likely were able to make better use of the supports within these comprehensive care models to maintain or regain workforce or educational accomplishment. In contrast to cross sectional measures of symptom severity that can be confounded by commonly erratic medication adherence, NLFET status provides a more sustained and ecologically relevant measure of functional capacity.

The disparity in functional outcomes based on DUP was also reflected in improved GAF scores at 6 and sustained at 12 months. While patients with longer DUP also experienced improvements within the first year of CSC, a persistent gap remained, underscoring the importance of early detection and echoing previous research on the importance of Low DUP for functional recovery and quality of life [4, 30, 31].

Regardless of the DUP cutoff used [13]—whether 90 days or the median DUP of 228 days —similar trends were observed, highlighting the importance of minimizing treatment delays both in absolute terms (less than 3 months) and relative to the sample (below the median split). The adverse impact of longer DUP on various psychometric measures of functioning diminished by the end of the first year of CSC, however, the effect on NLFET status persisted. In other words, although individuals with a longer DUP show improvement over time, they

continue to lag in labor force participation, which is crucial not only for their reintegration into society but also for the well-being of their families.

Our study has several limitations. As a secondary analysis, the findings should be considered exploratory and interpreted with caution, requiring further validation in future research. Re-examining existing data rather than testing a predefined hypothesis can introduce biases and limit generalizability. Moreover, multiple testing increases the risk of Type I errors, especially when numerous comparisons are made without proper adjustments [32]. Although we focused on theoretically driven comparisons to mitigate this risk, the results remain preliminary. Future studies should replicate these findings using rigorous statistical approaches that account for multiple testing.

Additionally, the follow-up period of one year might not be adequate to fully understand the impact of DUP reduction on functional outcomes. Longer-term studies, such as TIPS (The Early Treatment and Intervention in Psychosis Study), have shown sustained adverse effects of prolonged DUP over 5 and even 10 years after entry into care, suggesting that the benefits of early intervention may have been underestimated in this analysis. Also, the absolute levels of DUP in TIPS (5–10 weeks) were several-fold lower than in this sample. Lower absolute levels of DUP may be necessary to observe greater and more sustained improvements in functional outcomes. Also, the provision of care within best practice CSCs may have ameliorated the adverse impact of longer DUP on functioning. The effect of DUP may have been larger in less enriched clinical care models or newer CSCs, in samples with poorer social determinants, or in settings with more disadvantageous labor market conditions.

This study supports the significant impact of DUP on functional outcomes in individuals with first-episode psychosis, even those enrolled in relatively enriched specialty care services (i.e., CSCs) and evident within the first year of care. Patients with a shorter delay to care demonstrated better employment, schooling, and job-seeking outcomes, as well as better global functioning. These findings reinforce the importance of early detection (or systematic efforts to reduce DUP) as a necessary complement to CSC. The resulting comprehensive early intervention services can then aim to improve pathways *to* and *through* care within a population health framework that can deliver both broader (illness and functioning) and less disparate outcomes [33].

## Supporting information

**S1 Table. Differences in outcomes between Low vs High DUP groups.**
(DOCX)

**S1 Fig. Bureau of Labor Statistics linked categorization.**
(DOCX)

## Author Contributions

**Conceptualization:** Toni Gibbs-Dean, Cenk Tek, Matcheri S. Keshavan, Vinod H. Srihari.

**Data curation:** Hadar Hazan.

**Formal analysis:** Bin Zhou, Fangyong Li.

**Funding acquisition:** Cenk Tek, Matcheri S. Keshavan, Vinod H. Srihari.

**Investigation:** Vinod H. Srihari.

**Methodology:** Cenk Tek, Matcheri S. Keshavan, Vinod H. Srihari.

**Project administration:** Hadar Hazan.

**Supervision:** Fangyong Li, Cenk Tek, Matcheri S. Keshavan, Vinod H. Srihari.

**Validation:** Fangyong Li.

**Visualization:** Bin Zhou, Fangyong Li.

**Writing – original draft:** Hadar Hazan.

**Writing – review & editing:** Hadar Hazan, Sümeyra N. Tayfur, Bin Zhou, Fangyong Li, Toni Gibbs-Dean, Sneha Karmani, Emily Kline, Maria Ferrara, Silvia Corbera, Sarah Riley, Laura A. Yoviene Sykes, Cenk Tek, Vinod H. Srihari.

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
