## [Decision Letter · Decision Letter 0]

20 Mar 2024

PONE-D-23-42589Duration of Untreated Psychosis and Its Association with Clinical and Functional Outcomes: A 6 and 12-Month Follow-Up Study of First-Episode Psychosis Cohorts at Two LocationsPLOS ONE

Dear Dr. hazan,

Thank you for submitting your manuscript to PLOS ONE. After careful consideration, we feel that it has merit but does not fully meet PLOS ONE’s publication criteria as it currently stands. Therefore, we invite you to submit a revised version of the manuscript that addresses the points raised during the review process.

Both reviewers and also my own review, have highlighted certain methodological aspects that could enhance the clarity of your findings. Consequently, I kindly request that you thoroughly address each of the points raised by the reviewers and revise the pertinent sections of your manuscript accordingly.

We look forward to receiving your revised manuscript.

Kind regards,

Inga Schalinski

Academic Editor

PLOS ONE

**Journal requirements:**

2. In the online submission form, you indicated that your data will be submitted to a repository upon acceptance.  We strongly recommend all authors deposit their data before acceptance, as the process can be lengthy and hold up publication timelines. Please note that, though access restrictions are acceptable now, your entire minimal dataset will need to be made freely accessible if your manuscript is accepted for publication. This policy applies to all data except where public deposition would breach compliance with the protocol approved by your research ethics board. If you are unable to adhere to our open data policy, please kindly revise your statement to explain your reasoning and we will seek the editor's input on an exemption.  

5. We notice that your supplementary figures are uploaded with the file type 'Figure'. Please amend the file type to 'Supporting Information'. Please ensure that each Supporting Information file has a legend listed in the manuscript after the references list.

6. We notice that your supplementary [tables] are included in the manuscript file. Please remove them and upload them with the file type 'Supporting Information'. Please ensure that each Supporting Information file has a legend listed in the manuscript after the references list.

7. Please include captions for your Supporting Information files at the end of your manuscript, and update any in-text citations to match accordingly. Please see our Supporting Information guidelines for more information: http://journals.plos.org/plosone/s/supporting-information

Reviewers' comments:

Reviewer's Responses to Questions

**Comments to the Author**

1. Is the manuscript technically sound, and do the data support the conclusions?

Reviewer #1: Yes

Reviewer #2: Partly

2. Has the statistical analysis been performed appropriately and rigorously? 

Reviewer #1: No

Reviewer #2: No

3. Have the authors made all data underlying the findings in their manuscript fully available?

Reviewer #1: Yes

Reviewer #2: No

4. Is the manuscript presented in an intelligible fashion and written in standard English?

Reviewer #1: Yes

Reviewer #2: Yes

5. Review Comments to the Author

Reviewer #1: This is a 12-month follow-up study comparing symptom and functional outcomes, including non-NLFET status between high vs low DUP groups in FEP cohort in US at 6 and 12 months of follow-up. The results showed that low DUP group was associated with significantly greater improvement in global functioning and higher rate of non-NLFET status at follow-up than high DUP group across and within sites. This is generally consistent with a large body of literature indicating that prolonged / longer DUP is associated with poorer clinical and functional outcomes in FEP, at least in short-term follow-up. A number of methodological issues require further clarification:

1. Some important clinical variables of study sample were not reported and included in the comparison analyses (as covariates for adjustment), such as age at onset of psychosis and diagnostic categories (e.g. schizophrenia-spectrum vs. other non-affective psychoses, as the former may be associated with poorer outcome). Did the study consider to adjust for potential group difference in socio-demographic variables as stated in Table S1? This study included premorbid functioning but only measured childhood stage (7-11 yrs) in PAS, with the argument that adolescence period overlaps with onset of psychosis / prodrome. On the other hand, adolescence period in PAS is further divided into early and late adolescence (11-15y; 16-18y). In fact, most FEP studies include PAS childhood, early and late adolescence in the PAS assessment. To avoid overlap with onset of illness, these studies usually adopted either 6-month or more stringently 12-month window period (i.e., PAS measurement only up to 12 months prior to onset of psychosis to minimize overlap with prodrome period). Childhood PAS is too restrictive and is a limitation of the study.

2. The study did not report inter-rater reliability in outcome measurement. This is important, esp. this study was conducted across two sites.

3. Quality of life scale (QLS), strictly speaking, is a measure of psychosocial functioning (also include intrinsic motivation component in some items) rather than subjective quality of life. Hence there is overlap between GAF role and social and QLS measurement as functional outcome measures.

4. The study abstract reported that it adopted repeated measures ANOVAs for DUP group outcome comparison over time. However, in the method section, it stated that linear mixed effect models and GEE were used. They seem to be different in statistical approaches, and require further clarification.

5. The study used median split DUP to define low and high DUP. However, the median DUP is different in between-site DUP group comparison, and within-site (ED) DUP group comparison. A better approach would be to use cutoff of <3 months (as it is defined by early intervention guideline as the target for short DUP) as short vs long DUP.

6. In Table 1: high v low DUP (GF current score) at 6 months, the score of 33.9 (low DUP) vs. 47.5 (high DUP) seems to be at odds with the reported finding? the lower the GF score, the worse the global functioning.

Reviewer #2: This study from Hazan et al examined the relationship between DUP and NLFET status across two CSC clinics and within the STEP site which employed an ED strategy.

The authors found that individuals with a lower DUP were more likely to have a non-NLFET status both at 6 and 12 months after admission. GAF scores also improved in both high and low DUP groups with a larger change occurring in the low DUP group.

The study has several strengths including a relatively large cohort for a study of this complexity and phenotypic depth, attempts to control for potential confounds including pre-morbid functioning, IQ, and symptom burden. Additionally, the manuscript provides a concise review of the DUP literature as it relates to functional status.

Unfortunately, I have several main concerns with the manuscript surrounding readability and the overall presentation of the data and analyses that make interpretation of the results challenging.

At times, the data and findings are presented in a confusing manner and would benefit from some clarifications and a consideration different visualizations/analyses. For example, the following paragraph:

“At baseline, Individuals with high or low DUP exhibited similar odds of not being in the

labor force or education (NLFET) (61% vs. 70.7%, odds ratio = 0.81, p = 0.45). At 6 months

both groups experienced increased engagement in vocational activities at 6 months (high DUP:

75.1%, low DUP: 89.1%), but the improvement was significantly greater in the low DUP group

(odds ratio: 0.45, p = 0.03). By 12 months, the low DUP group had significantly higher odds of

not being in NLFET (90.4%) compared to the high DUP group (69.5%). This translates to a larger increase in vocational engagement/education (7.32 vs. 2.16, odds ratio 0.29, p = 0.003).”

The first sentence makes it seem that 61% of high and 70.7% of low DUP individuals had NLFET status (i.e., “not being in the labor force of education). However, the next sentence says that they had increased engagement with 75.1% of high and 89.1% of low DUP engaged in work. I believe that this confusion is due to a typo in the first sentence that should read similar odds of not being in the NLFET status. Otherwise, this would imply that 70% of low DUP individuals were initially not working and then 89% were working.

Assuming this typo, I am curious about the underlying statistics that lead to a significant p-value of 0.03. It appears that there was an absolute increase in high DUP employment of 14.1% (61% to 75.1%) and 18.4% in the low DUP employment group (70.7% to 89.1%) leading to a small absolute difference of 4.3% in NLFET status between these two groups. The confidence intervals of these two data points are clearly overlapping in Figure 2a. They only barely do not overlap at 12 months and this difference would not likely survive multiple testing correction. Additionally, the significant effect of DUP length on NFLET status seems to go away (confidence intervals completely overlapping) in Figure 2b when compared within the STEP group.

In the manuscript and Table 1, it is not clear what the values in the NLFET column represent. I deduced that they were odds ratios by looking at additional figures but this should be made clear. How was the increase in vocational employment quantified in this comparison? “7.32 vs. 2.16, odds ratio 0.29, p = 0.003”

Concerningly, in all analyses, there is no mention of how many participants fall into the high and low DUP, respectively. Although the authors mention a median split of DUP, depending on the distribution of DUP lengths in the cohorts, there may be a significant imbalance in the number of low vs. high DUP participants. I would request that the numbers underlying each analysis be shown so that the readers can see how many individuals are comprising each subgroup.

The authors interchangeably present raw values with change in values (PANSS). There are likely errors and typos across tables and figures. For example, Table 1 has identical values for GF Current, Role, Social, QoL, Pos/Neg/Gen/Total PANSS between 12 month high and low DUP Across Sites and within STEP ED. The chances of these values all being identical across individuals is essentially zero and likely represents a typographical error. To this point, the Total PANSS score in Figure S3 appears substantially different than S4.

Given the lack of clearly presented data, typographical errors, and minimally significant results that would not survive multiple testing correction, I would need to see substantial improvements to the manuscripts readability and presentation of the underlying data before recommending acceptance for publication.

6. PLOS authors have the option to publish the peer review history of their article (what does this mean?). If published, this will include your full peer review and any attached files.

Reviewer #1: No

Reviewer #2: No

---

## [Author Response · Author response to Decision Letter 0]

20 Sep 2024

We appreciate the opportunity to respond to these careful reviews and believe this has helped us to improve the quality of the manuscript. We have below quoted concerns raised by the reviewers and responded to these concerns in detail and where appropriate have described changes made in the revised submission.

Reviewer #1: 

• Some important clinical variables of study sample were not reported and included in the comparison analyses (as covariates for adjustment), such as age at onset of psychosis and diagnostic categories (e.g. schizophrenia-spectrum vs. other non-affective psychoses, as the former may be associated with poorer outcome). Did the study consider to adjust for potential group difference in socio-demographic variables as stated in Table S1?

Response: Thank you for your question. DUP or delay in accessing treatment is downstream, or the result of a combination of socio-demographic and clinical risk factors. We thus view DUP as a latent trait when assessing its moderation of longitudinal outcomes of treatment. In other words, the reason that a patient was categorized into high DUP or low DUP group was a consequence of many risk factors, like some variables in Table S1. The purpose of this study is to assess the moderation effect of DUP as a proxy of all these factors. Including these in the model may cancel out the effect of DUP itself. 

• This study included premorbid functioning but only measured childhood stage (7-11 yrs) in PAS, with the argument that adolescence period overlaps with onset of psychosis / prodrome. On the other hand, adolescence period in PAS is further divided into early and late adolescence (11-15y; 16-18y). In fact, most FEP studies include PAS childhood, early and late adolescence in the PAS assessment. To avoid overlap with onset of illness, these studies usually adopted either 6-month or more stringently 12-month window period (i.e., PAS measurement only up to 12 months prior to onset of psychosis to minimize overlap with prodrome period). Childhood PAS is too restrictive and is a limitation of the study.

Response: We appreciate the reviewer’s insightful comment. While we acknowledge that measuring childhood functioning in the PAS is a limitation, our deliberate focus on this stage was driven by the need to minimize overlap with the prodromal period that can precede psychosis onset by several months to even years. This approach aligns with practices adopted by the first and only other successful DUP reduction effort (Joa at al., 2008; Schizophrenia bulletin, 34(3), 466-472)

• The study did not report inter-rater reliability in outcome measurement. This is important, esp. this study was conducted across two sites.

Response: Thank you for this comment. We would like to clarify that raters at each site were indeed trained using a set of common vignettes. These vignettes were developed from historical cases at each site. The focus of the reliability assessment was on diagnosis, Global Assessment of Functioning (GAF) from the Structured Interview for Psychosis-risk Syndromes (SIPS), and Duration of Untreated Psychosis (DUP). After achieving adequate reliability, raters repeated this exercise after every 20 new subjects that were recruited at each site. This was done to ensure ongoing monitoring of reliability. Furthermore, annual meetings between the sites were conducted to reduce drift and increase reliability. We have included this information in our revised manuscript (p. 5).

• Quality of life scale (QLS), strictly speaking, is a measure of psychosocial functioning (also include intrinsic motivation component in some items) rather than subjective quality of life. Hence there is overlap between GAF role and social and QLS measurement as functional outcome measures.

Response: We appreciate this comment. Indeed, our decision to incorporate both (overlapping) measures was driven by our aim to provide a more comprehensive view of functioning. Additionally, we believe that this approach would facilitate comparisons with other studies that have examined similar variables. We hope this clarifies our rationale.

• The study abstract reported that it adopted repeated measures ANOVAs for DUP group outcome comparison over time. However, the method section, it stated that linear mixed effect models and GEE were used. They seem to be different in statistical approaches, and require further clarification.

Response: Thank you for pointing this out. This is an error, and we have now revised the abstract and edited and clarified the description of the statistical analysis in the Methods section (p.6): 

Classification into Low vs High DUP was based on (i) a median split of the distribution and (ii) the World Health Organization (WHO)/ International Early Psychosis Association (IEPA) aspirational standard of 12 weeks25. Linear Mixed Model Repeated measures (MMRM) analysis was utilized to evaluate change in continuous outcomes (functioning, symptom severity, quality of life) and a Generalized Estimating Equation (GEE) multivariate model was utilized to evaluate change in binary outcomes (for employment, being in school, or job hunting), in which visit times (baseline, 6-month and 12-month) were adjusted as covariates for all outcomes, with an unstructured variance-covariance matrix specified to account for within-subject correlation of repeatedly measured values during periods. To quantify the sizes of estimated effects, least square means for continuous outcomes, odds ratios for binary outcomes, and confidence intervals (CI) were reported at each time point by high and low DUP.

Also, we have edited the figures for easier interpretability.

• The study used median split DUP to define low and high DUP. However, the median DUP is different in between-site DUP group comparison, and within-site (ED) DUP group comparison. A better approach would be to use cutoff of <3 months (as it is defined by early intervention guideline as the target for short DUP) as short vs long DUP.

Response: Thank you for this valuable suggestion. In response to your feedback, we have restructured the manuscript to enhance clarity and consistency. We removed the between-site comparisons and focused on presenting results for the same sample using both a median split and a 3-month cutoff for defining short vs. long DUP. This approach aligns with early intervention guidelines and provides a more standardized analysis. Importantly, the reanalysis using the 3-month cutoff did not significantly alter the overall findings of the study, confirming the robustness of our results. These changes have been applied throughout the manuscript, as indicated.

• In Table 1: high v low DUP (GF current score) at 6 months, the score of 33.9 (low DUP) vs. 47.5 (high DUP) seems to be at odds with the reported finding? the lower the GF score, the worse the global functioning.

Response: Thank you for your careful review and for highlighting this discrepancy. We apologize for any confusion caused by the initial presentation of the data. Upon reviewing the table, we recognized that the presentation could be misleading due to the organization of the group names. To improve clarity, we have moved the original table to the appendix and replaced it with figures that clearly illustrate the outcomes for each group based on both the median split and the 3-month cutoff for DUP. These figures provide a more straightforward visualization of the data and align with the reported findings in the text. We believe these changes enhance the manuscript's readability and help avoid potential misunderstandings.

Reviewer #2:

• This study from Hazan et al examined the relationship between DUP and NLFET status across two CSC clinics and within the STEP site which employed an ED strategy. At times, the data and findings are presented in a confusing manner and would benefit from some clarifications and a consideration different visualizations/analyses. For example, the following paragraph:

“At baseline, Individuals with high or low DUP exhibited similar odds of not being in the labor force or education (NLFET) (61% vs. 70.7%, odds ratio = 0.81, p = 0.45). At 6 months both groups experienced increased engagement in vocational activities at 6 months (high DUP: 75.1%, low DUP: 89.1%), but the improvement was significantly greater in the low DUP group (odds ratio: 0.45, p = 0.03). By 12 months, the low DUP group had significantly higher odds of not being in NLFET (90.4%) compared to the high DUP group (69.5%). This translates to a larger increase in vocational engagement/education (7.32 vs. 2.16, odds ratio 0.29, p = 0.003).” 

The first sentence makes it seem that 61% of high and 70.7% of low DUP individuals had NLFET status (i.e., “not being in the labor force of education). However, the next sentence says that they had increased engagement with 75.1% of high and 89.1% of low DUP engaged in work. I believe that this confusion is due to a typo in the first sentence that should read similar odds of not being in the NLFET status. Otherwise, this would imply that 70% of low DUP individuals were initially not working and then 89% were working.

Response: We apologize for the confusion caused by the presentation of our data on the NLFET status. You’re correct in noting that there seems to be a discrepancy in the first sentence. The sentence should indeed read: “At baseline, individuals with high or low DUP exhibited similar odds of being in the NLFET status.”

However, in the revised paper, we changed the wording and reversed our presentation to make it more readable. Instead of referring to “non-NLFET”, we refer to NLFET throughout the paper. We also revised the figures for this outcome (p.7, p. 11)

: 

“At baseline, no significant differences were found in NLFET status between high and low DUP groups in either the 3-month cut-off (difference: 0.07, p = 0.40) or the median split model (difference: 0.05, p = 0.45). By 6 months, the low DUP group showed significantly greater improvement in NLFET status (3-month cut-off: -0.16, p = 0.03; median split: -0.13, p = 0.03). At 12 months, this trend persisted, with the low DUP group continuing to demonstrate a reduction in the number of participants meeting NLFET status (3-month cut-off: -0.18, p = 0.03; median split: -0.20, p = 0.003). This trend is visualized in Figure 1”

• Assuming this typo, I am curious about the underlying statistics that lead to a significant p-value of 0.03. It appears that there was an absolute increase in high DUP employment of 14.1% (61% to 75.1%) and 18.4% in the low DUP employment group (70.7% to 89.1%) leading to a small absolute difference of 4.3% in NLFET status between these two groups. The confidence intervals of these two data points are clearly overlapping in Figure 2a. They only barely do not overlap at 12 months and this difference would not likely survive multiple testing correction. Additionally, the significant effect of DUP length on NFLET status seems to go away (confidence intervals completely overlapping) in Figure 2b when compared within the STEP group.

Response: We appreciate your observation regarding the figures. The figures originally depicted odds, while we reported the significance of the odds ratio (High DUP / Low DUP). This led to visual overlap despite the presence of a significant difference. To address this, we have revised our approach and created two new figures that represent probabilities instead of odds (see above). We believe this change will provide a clearer visual representation of our findings. Please refer to the updated figures above. The odds ratios using 3-month DUP cut-off were 0.76 (95% CI: 0.53, 0.91) at 6 month follow-up, and 0.78 (0.52, 0.92) at 12 month. If using median-spilt, the ORs were 0.69 (0.52, 0.82) and 0.77 (0.60, 0.88) respectively. Both split methods showed similar moderate effect sizes in OR, which didn’t overlap with 1. Multiple correction is not the only way to control for type 1 error. The results of the two methods supported each other, indicating the robustness of our conclusion. We hope this addresses your concern and enhances the clarity of our report. 

• In the manuscript and Table 1, it is not clear what the values in the NLFET column represent. I deduced that they were odds ratios by looking at additional figures but this should be made clear. How was the increase in vocational employment quantified in this comparison? “7.32 vs. 2.16, odds ratio 0.29, p = 0.003”. 

Response: 

Thank you for your insightful feedback. We understand the confusion regarding the values in the NLFET column and have made several changes to improve clarity:

o Relocation of Table 1 to the Appendix: Table 1 has been relocated to the appendix for comprehensive reference, where it now includes more detailed descriptions to clarify the statistical representation of the data.

o Revised Presentation of NLFET Status: We have revised the manuscript to better articulate how NLFET status and changes in vocational employment were quantified. Specifically, we have clarified that the values represent odds ratios derived from logistic regression models, highlighting the changes in employment status between high and low DUP groups over time. These revisions are intended to enhance readability and ensure the data is accurately interpreted (see revised sections and figures above).

• Concerningly, in all analyses, there is no mention of how many participants fall into the high and low DUP, respectively. Although the authors mention a median split of DUP, depending on the distribution of DUP lengths in the cohorts, there may be a significant imbalance in the number of low vs. high DUP participants. I would request that the numbers underlying each analysis be shown so that the readers can see how many individuals are comprising each subgroup: 

Response: Thank you for raising this important point regarding the potential imbalance in the number of low vs. high DUP participants. To address this concern, we have restructured the manuscript to ensure clarity and consistency. The manuscript now focuses on a single sample size, which is divided either by a median split or a 3-month cutoff, as recommended. The number of participants in each group may vary depending on the time of follow-up and the outcome variable. These variations are clearly indicated in Table 1s. This approach allows us to transparently present the distribution of participants across different follow-up periods, ensuring the robustness of our analysis. We hope this clarifies our methodology and addresses your concern.. 

• The authors interchangeably present raw values with change in values (PANSS). There are likely errors and typos across tables and figures. For example, Table 1 has identical values for GF Current, Role, Social, QoL, Pos/Neg/Gen/Total PANSS between 12 month high and low DUP Across Sites and within STEP ED. The chances of these values all being identical across individuals is essentially zero and likely represents a typographical error. To this point, the Total PANSS score in Figure S3 appears substantially different than S4: 

Response: We acknowledge your point about the interchangeable presentation of raw values with changes in values. We revised our tables and figures to ensure consistency and accuracy in the presentation of our data. 

• Given the lack of clearly presented data, typographical errors, and minimally significant results that would not survive multiple testing correction, I would need to see substantial improvements to the manuscripts readability and presentation of the underlying data before recommending acceptance for publication

 Response: we revised our manuscript to ensure that our data was presented clearly and understandably. We replaced the presentation in the table with figures to enhance the interpretability of our results. We thoroughly proofread our manuscript to correct any typographical errors and improve its readability. We appreciate the time and effort spent in reviewing our work. The feedback was invaluable in helping us improve our manuscript. Necessary improvements were made and the manuscript was resubmitted for consideration. We were grateful for the constructive criticism received.

---

## [Decision Letter · Decision Letter 1]

6 Oct 2024

PONE-D-23-42589R1The Impact of Duration of Untreated Psychosis on Functioning and Quality of Life Over One Year of Coordinated Specialty Care (CSC)PLOS ONE

Dear Dr. hazan,

Thank you for submitting your manuscript to PLOS ONE. After careful consideration, we feel that it has merit but does not fully meet PLOS ONE’s publication criteria as it currently stands. Therefore, we invite you to submit a revised version of the manuscript that addresses the points raised during the review process. Reviewer one and I have concluded that the manuscript requires further revision. The comments need to be addressed that the reviewer has highlighted.

We look forward to receiving your revised manuscript.

Kind regards,

Inga Schalinski

Academic Editor

PLOS ONE

Journal Requirements:

Additional Editor Comments:

Consistency of abbreviations in Figure captions (e.g., 3m and 3mo); name all abbreviations in Figure caption

Reviewers' comments:

Reviewer's Responses to Questions

**Comments to the Author**

1. If the authors have adequately addressed your comments raised in a previous round of review and you feel that this manuscript is now acceptable for publication, you may indicate that here to bypass the “Comments to the Author” section, enter your conflict of interest statement in the “Confidential to Editor” section, and submit your "Accept" recommendation.

Reviewer #2: (No Response)

2. Is the manuscript technically sound, and do the data support the conclusions?

Reviewer #2: Partly

3. Has the statistical analysis been performed appropriately and rigorously? 

Reviewer #2: No

4. Have the authors made all data underlying the findings in their manuscript fully available?

Reviewer #2: No

5. Is the manuscript presented in an intelligible fashion and written in standard English?

Reviewer #2: Yes

6. Review Comments to the Author

Reviewer #2: In the revised manuscript, the authors have changed some components of the analysis and framing of the manuscript, however, my central issues regarding the fundamental significance of the findings and transparency of the data for the primary outcome remain.

First, the authors state that the confidence intervals for the odds ratios "did not overlap with 1," suggesting significance. However, this interpretation does not fully address the central issue when testing for a difference between two groups (high DUP vs. low DUP). Specifically, the correct statistical approach for testing whether there is a significant difference between two groups requires that their respective confidence intervals do not overlap with each other, not just that they don't overlap with 1.

While an odds ratio's confidence interval that does not overlap with 1 indicates a statistically significant association within a single group, this is not sufficient to conclude that the two groups (high vs. low DUP) are significantly different from each other. Overlap of the CIs between the two groups would suggest that any observed differences might be due to chance, even if the odds ratios for each group individually suggest significance. Switching from an odds ratio to probabilities does not address this concern and it is important to visualize the confidence intervals when presenting this data to give the readers a sense of the confidence of the results.

Thus, I remain concerned that the current explanation of the findings does not address whether the confidence intervals for the high and low DUP groups overlap with each other, which is essential for confirming a significant difference between the two groups. Without clarification on this point, the statistical argument for a significant difference between high and low DUP groups remains unclear.

The authors noted that "multiple correction is not the only way to control for type I error" but do not mention any controls for type I error in the manuscript or correction for multiple comparisons. Given the dozens of statistical tests and comparisons made in the manuscript and a threshold level significance (p = 0.03), strong justification should be provided for why multiple comparison was not deemed necessary. The consistency of the results between median split vs. 3-month cutoff is not sufficient, especially given that we do not know how the samples differ with respect to the number of participants in high vs. low DUP with each split.

Another significant issue that remains unresolved is the lack of clearly reported group sizes for the high DUP and low DUP groups. In the initial review, I raised concerns about potential imbalances between these two groups and how this could affect the robustness of the comparisons.

The authors state: “The manuscript now focuses on a single sample size, which is divided either by a median split or a 3-month cutoff, as recommended.”

I am not sure what focusing on a “single sample size” means. Perhaps the authors mean that they grouped the cohorts together and then split based on median DUP?

The authors then state, “The number of participants in each group may vary depending on the time of follow-up and the outcome variable. These variations are clearly indicated in Table 1s.”

Even if the number of participants in each group vary depending on the follow-up and outcome variable, they sample size (n) of high and low DUP groups should still be presented with each figure. Despite the authors stating that “these variations are clearly indicated in Table 1s,” table Table 1s does not provide any information regarding the size of the high vs. low DUP sample or baseline demographic and clinical characteristics of each of these groups.

This is essential information for assessing whether any differences observed between the groups could be due to imbalances in sample size. Without explicit reporting of the group sizes, it remains difficult to evaluate the robustness of findings to ensure that they are not simply due to chance.

7. PLOS authors have the option to publish the peer review history of their article (what does this mean?). If published, this will include your full peer review and any attached files.

Reviewer #2: No

---

## [Author Response · Author response to Decision Letter 1]

10 Oct 2024

We appreciate the thoughtful comments and the opportunity to address the concerns raised. Below we provide detailed responses to each of the points highlighted by the reviewer:

Reviewer #2: In the revised manuscript, the authors have changed some components of the analysis and framing of the manuscript, however, my central issues regarding the fundamental significance of the findings and transparency of the data for the primary outcome remain.

First, the authors state that the confidence intervals for the odds ratios "did not overlap with 1," suggesting significance. However, this interpretation does not fully address the central issue when testing for a difference between two groups (high DUP vs. low DUP). Specifically, the correct statistical approach for testing whether there is a significant difference between two groups requires that their respective confidence intervals do not overlap with each other, not just that they don't overlap with 1.

While an odds ratio's confidence interval that does not overlap with 1 indicates a statistically significant association within a single group, this is not sufficient to conclude that the two groups (high vs. low DUP) are significantly different from each other. Overlap of the CIs between the two groups would suggest that any observed differences might be due to chance, even if the odds ratios for each group individually suggest significance. Switching from an odds ratio to probabilities does not address this concern and it is important to visualize the confidence intervals when presenting this data to give the readers a sense of the confidence of the results.

Response: We respectfully disagree with the reviewer’s interpretation of the Odds Ratio (OR) analysis. An Odds Ratio quantifies the magnitude of an association between a binary outcome and an exposure. When comparing two dichotomized groups (e.g., High DUP vs. Low DUP), the OR measures the relative odds of the outcome occurring in one group compared to the other.

In our analysis of NLFET status, we examined the odds of NLFET between two groups (High vs. Low DUP). The reported Odds Ratio provides a legitimate measure for comparing NLFET status across these groups, rather than being an assessment limited to a single group.

For example, we stated:

"By 6 months, the High DUP group demonstrated significantly higher odds of NLFET status compared to the Low DUP group in both 3-month (OR = 3.25, 95% CI: 1.11 to 9.54, p = 0.03)…"

This indicates that the odds of NLFET were 3.25 times higher in the High DUP group compared to the Low DUP group. This analysis indeed reflects a comparison between the two groups. Thus, the reviewer’s comment that "an Odds Ratio’s confidence interval not overlapping with 1 indicates a statistically significant association within a single group, but does not establish a significant difference between groups" appears to conflate different aspects of OR interpretation.

There is no concept of an "Odds Ratio for each group" in our analysis. The Odds Ratio represents the ratio of odds between the groups, calculated based on the odds within each group. This approach aligns with standard epidemiological practice for group comparisons using ORs.

Reviewer:

Thus, I remain concerned that the current explanation of the findings does not address whether the confidence intervals for the high and low DUP groups overlap with each other, which is essential for confirming a significant difference between the two groups. Without clarification on this point, the statistical argument for a significant difference between high and low DUP groups remains unclear.

Response: We respectfully disagree with the reviewer’s interpretation regarding confidence intervals (CIs). The confidence intervals for the groups generated by each of the cutoffs do not overlap each other’s point estimates and thus the reported intervals are consistent with our inference about traditional statistical significance. In other words, the mere overlapping of confidence intervals is not the issue.

It’s a common misconception that the statistical significance must come with non-overlapping confidence interval. Please see this highly cited (> 3000 citations) reference paper: 

Greenland S, Senn SJ, Rothman KJ, Carlin JB, Poole C, Goodman SN, Altman DG. Statistical tests, P values, confidence intervals, and power: a guide to misinterpretations. Eur J Epidemiol. 2016 Apr;31(4):337-50. doi: 10.1007/s10654-016-0149-3. Epub 2016 May 21. PMID: 27209009; PMCID: PMC4877414. 

In scenario 21, the authors explained very well why it is wrong to state “If two confidence intervals overlap, the difference between two estimates or studies is not significant”. This paper provides a thorough explanation of why this reasoning is flawed.

As explained above, the 95% CI of odds ratio without including 1 is consistent with a p value < 0.05, indicating the significant difference in odds of NLFET between the two groups. To better illustrate the group differences, we plotted the predicted probability (converted from the estimated odds through the reverse link function) of NLFET instead of log odds (logit) which was used in the previous version. The predicted probability is very close to the observed proportion of NLFET in each group at different time points and is commonly used for logistic regression or GEE models. 

Reviewer: 

The authors noted that "multiple correction is not the only way to control for type I error" but do not mention any controls for type I error in the manuscript or correction for multiple comparisons. Given the dozens of statistical tests and comparisons made in the manuscript and a threshold level significance (p = 0.03), strong justification should be provided for why multiple comparison was not deemed necessary. The consistency of the results between median split vs. 3-month cutoff is not sufficient, especially given that we do not know how the samples differ concerning the number of participants in high vs. low DUP with each split.

Response: Thank you for your comment. In response to Reviewer 1's suggestion, we used both the 3-month cutoff and the median split to define the high/low DUP groups, demonstrating the robustness of the findings across different criteria. Both approaches represent non-arbitrary ways to establish thresholds: the median split is a common practice and was used in the largest U.S. study on DUP, while the 3-month cutoff is a well-recognized benchmark endorsed by the World Health Organization (WHO).

The consistency of results across these thresholds strengthens the conclusion that the observed differences between the high and low DUP groups are robust. As noted in the manuscript, this analysis was secondary and exploratory in nature. Further validation in future studies is warranted to confirm these findings.

We acknowledge that this was a secondary analysis and should be regarded as exploratory, thus requiring further validation in future studies. To address this point, we added a sentence in the Discussion section (p. 12) to explicitly clarify that this analysis is exploratory in nature, acknowledge its limitations, and cite the reference by Bender and Lange (2001) on multiple testing considerations (Journal of Clinical Epidemiology, Volume 54, Issue 4, Pages 343-349, https://doi.org/10.1016/S0895-4356(00)00314-0): 

“Our study has several limitations. As a secondary analysis, the findings should be considered exploratory and interpreted with caution, requiring further validation in future research. Re-examining existing data rather than testing a predefined hypothesis can introduce biases and limit generalizability. Moreover, multiple testing increases the risk of Type I errors, especially when numerous comparisons are made without proper adjustments33. Although we focused on theoretically driven comparisons to mitigate this risk, the results remain preliminary. Future studies should replicate these findings using rigorous statistical approaches that account for multiple testing” (page 12). 

Reviewer:

Another significant issue that remains unresolved is the lack of clearly reported group sizes for the high DUP and low DUP groups. In the initial review, I raised concerns about potential imbalances between these two groups and how this could affect the robustness of the comparisons.

Response: Thank you for the comments (p. 6 statistical analysis section). We have added the group sizes to the manuscript. For the median split cutoff, both groups have the same sample size (High DUP, n = 123; Low DUP, n = 123). For the 3-month cutoff following the WHO/IEPA aspirational standard of 12 weeks, there is an imbalance in sample size (High DUP, n = 173; Low DUP, n = 73), but this did not affect the results or conclusions

Reviewer:

The authors state: “The manuscript now focuses on a single sample size, which is divided either by a median split or a 3-month cutoff, as recommended.” I am not sure what focusing on a “single sample size” means. Perhaps the authors mean that they grouped the cohorts together and then split based on median DUP?

Response: We apologize for the confusion. This phrase refers to the original submission, in which analyses were separated by site. We have since revised this approach based on the reviewers' feedback. The phrase in the response letter is a typographical error. 

Reviewer:

The authors then state, “The number of participants in each group may vary depending on the time of follow-up and the outcome variable. These variations are clearly indicated in Table 1s.” 

Even if the number of participants in each group vary depending on the follow-up and outcome variable, they sample size (n) of high and low DUP groups should still be presented with each figure. Despite the authors stating that “these variations are clearly indicated in Table 1s,” Table 1s does not provide any information regarding the size of the high vs. low DUP sample or baseline demographic and clinical characteristics of each of these groups.

This is essential information for assessing whether any differences observed between the groups could be due to imbalances in sample size. Without explicit reporting of the group sizes, it remains difficult to evaluate the robustness of findings to ensure that they are not simply due to chance.

Response: As noted in a similar suggestion above, we have now inserted the sizes of the groups created by each of the two cutoff criteria (p. 6 statistical analysis section). To the second point, about these group size imbalances affecting the results, we have clarified that this was not the case with our data. 

Editor Comments. 

Consistency of abbreviations in Figure captions (e.g., 3m and 3mo); name all abbreviations in Figure caption. 

Response: We have standardized all abbreviations to "3mo" for consistency and ensured that all abbreviations are fully defined in each figure caption. These changes have been made in the revised manuscript.

---

## [Editor Report · Decision Letter 2]

14 Oct 2024

The Impact of Duration of Untreated Psychosis on Functioning and Quality of Life Over One Year of Coordinated Specialty Care (CSC)

PONE-D-23-42589R2

Dear Dr. hazan,

We’re pleased to inform you that your manuscript has been judged scientifically suitable for publication and will be formally accepted for publication once it meets all outstanding technical requirements.

Kind regards,

Inga Schalinski

Academic Editor

PLOS ONE
---

## [Editor Report · Acceptance letter]

5 Nov 2024

PONE-D-23-42589R2 

PLOS ONE

Dear Dr. Hazan, 

I'm pleased to inform you that your manuscript has been deemed suitable for publication in PLOS ONE. Congratulations! Your manuscript is now being handed over to our production team.

Kind regards, 

on behalf of

Dr. Inga Schalinski 

Academic Editor

PLOS ONE